# Exploring the Psychological Impacts of COVID-19 Social Restrictions on International University Students: A Qualitative Study

**DOI:** 10.3390/ijerph19137631

**Published:** 2022-06-22

**Authors:** Amani Al-Oraibi, Lauren Fothergill, Mehmet Yildirim, Holly Knight, Sophie Carlisle, Mórna O’Connor, Lydia Briggs, Joanne R. Morling, Jessica Corner, Jonathan K. Ball, Chris Denning, Kavita Vedhara, Holly Blake

**Affiliations:** 1School of Medicine, University of Nottingham, Nottingham NG7 2UH, UK; msaaa28@exmail.nottingham.ac.uk (A.A.-O.); holly.knight@nottingham.ac.uk (H.K.); sophie.r.carlisle@kcl.ac.uk (S.C.); joanne.morling@nottingham.ac.uk (J.R.M.); kavita.vedhara@nottingham.ac.uk (K.V.); 2Department of Respiratory Sciences, University of Leicester, Leicester LE1 9HN, UK; 3Division of Health Research, Lancaster University, Lancaster LA1 4AG, UK; msalf5@exmail.nottingham.ac.uk; 4School of Health Sciences, University of Nottingham, Nottingham NG7 2HA, UK; mehmet.yildirim@nottingham.ac.uk (M.Y.); msamo8@exmail.nottingham.ac.uk (M.O.); lydia.briggs@tgh.nhs.uk (L.B.); 5NIHR Nottingham Biomedical Research Centre, Nottingham NG7 2UH, UK; 6University Executive Board, University of Nottingham, Nottingham NG7 2RD, UK; jessica.corner@nottingham.ac.uk; 7School of Life Sciences, University of Nottingham, Nottingham NG7 2UH, UK; jonathan.ball@nottingham.ac.uk (J.K.B.); chris.denning@nottingham.ac.uk (C.D.); 8Biodiscovery Institute, University of Nottingham, Nottingham NG7 2RD, UK

**Keywords:** COVID-19, higher education, university, mental health, well-being, social isolation, students

## Abstract

The global COVID-19 pandemic has impacted on the mental well-being of university students, but little attention has been given to international students, who may have a unique experience and perspective. The aim of this study was to explore the views of international students and university staff towards COVID-19 restrictions, self-isolation, their well-being, and support needs, through eight online focus groups with international students (*n* = 29) and semi-structured interviews with university staff (*n* = 17) at a higher education institution in England. Data were analysed using an inductive thematic approach, revealing three key themes and six subthemes: (1) practical, academic, and psychological challenges faced during self-isolation and the COVID-19 pandemic; (2) coping strategies to self-isolation and life during the pandemic; and (3) views on further support needed for international students. International students faced practical, academic, and psychological challenges during the COVID-19 pandemic, particularly relating to the rapid transition to online learning and the impact of social restrictions on integration with peers and well-being. Online social connections with peers, family, or new acquaintances reduced feelings of isolation and encouraged involvement in university life. Despite raising mental health concerns, most international students did not access mental health support services. Staff related this to perceived stigma around mental health in certain cultural groups. In conclusion, international students experienced specific practical and emotional challenges during the pandemic, and are at risk of mental ill-health, but may not actively seek out support from university services. Proactive and personalised approaches to student support will be important for positive student experiences and the retention of students who are studying abroad in the UK higher education system.

## 1. Introduction

Since the outbreak of Coronavirus disease (COVID-19) in December 2019 in China, Severe Acute Respiratory Syndrome Coronavirus-2 (SARS-CoV-2) has spread rapidly to almost all parts of the world, infecting over 219 million people and claiming nearly five million lives up until October 2021 [1]. In the United Kingdom (UK), in March 2020, the World Health Organization (WHO) declared the situation a global pandemic [2]. In response to the rising number of cases and deaths attributed to COVID-19, the UK government implemented necessary restrictive measures to contain the spread of the virus, including stay-at-home orders, self-isolation measures, social distancing, and various levels of lockdowns and border closures [3]. Following national guidelines, universities around the world locked down their campuses to reduce the spread of the virus, and shifted their operations to online learning [4].

Though the risk of serious health implications following infection is lower in younger populations [5], the mental health impacts of living through a pandemic are now established, both in the general population [6,7] and in younger populations [8,9], especially in university and college students [10,11]. Student populations have faced considerable disruption to their academic and social lives, including the rapid transition to remote working, new online assessment methods, changes to workloads and performance expectations, closed university facilities and residences, reduced social interaction, financial concerns related to lost part-time jobs, and worries about future education and careers [12].

Studies are rapidly emerging which explore the psychological impacts of the COVID-19 pandemic on university students [4,10,11,13]. Students have faced considerable emotional challenges following COVID-19 restrictions, such as feelings of loneliness, stress, low mood, anxiety, and depression [14,15]. Various COVID-19 social restrictions (e.g., periods of national lockdown or self-isolation, defined as a period of remaining apart from others to prevent the transmission or acquisition of an infectious disease, or social distancing of >2m apart) have led to lifestyle behaviour changes, with increases in sedentary behaviour and reduced sleep or sleep quality, which are known risk factors for decreased well-being [8,9,10]. Despite a plethora of studies exploring the impacts of the pandemic on higher education students, there has been little attention to the specific experiences of international students, which may differ to those of their non-international peers. According to the United Nations Educational, Scientific and Cultural Organization (UNESCO) definition, international (or internationally mobile) students are those who have crossed a national or territorial border for the purpose of education and are now enrolled outside their country of origin [16].

International students already face significant challenges when studying abroad, related to adapting to new cultures, dealing with language barriers, homesickness, and financial issues [17]. The COVID-19 social restrictions may have exacerbated these existing challenges for international students, potentially leaving them more vulnerable to inequalities or mental health concerns. Since the pandemic began, prospective international students have been more likely to make major changes to their education plans, such as postponing university enrolment, and enrolling in a home country institution, rather than travelling abroad for study [18]. Despite this, many international students were already enrolled in, or arrived at, university in the UK during the midst of the pandemic. International students constitute a high proportion of the domestic higher education market. In 2019/20, there were 538,615 international students studying in the UK (142,985 from the European Union (EU) and 395,630 non-EU) [19]. Yet, this figure is declining, with an estimated 47% decrease in international student enrolments in the next academic year, costing the sector £2.5 billion [20].

The retention of existing international students, and attraction of new students will be paramount for the future sustainability of global academic institutions. Further, research has identified a need to explore the concerns and experiences of international students during the COVID-19 outbreak [3,21,22]. According to the Student Covid Insights Survey (SCIS) conducted by the Office for National Statistics (ONS) in 2020, the COVID-19 pandemic has affected the mental health and well-being of over 50% of students in higher education in England [23]. The SCIS survey also shows that students showed lower levels of life satisfaction and happiness, and higher levels of anxiety during the pandemic than the general population. Similarly, a study that looked at the mental health impacts on international university students in both the UK and the United States of America (USA) found that almost 85% of all international students had moderate-to-high perceived stress from COVID-19-related stressors, such as lack of social support [24]. Furthermore, approximately 18% had moderate-to-severe symptoms of insomnia, and 12% had moderate-to-severe anxiety and depression symptoms [24]. These issues are exacerbated by multiple periods of self-isolation that many international students have experienced, which has ultimately been very challenging for their mental well-being [10]. Although knowledge in this field is emerging, there is limited published literature on the experiences of international students who are registered for study in the UK, with relation to social restrictions and well-being.

The University Mental Health Charter [25], created in 2019, provides guidance on making mental health a priority in universities. It advocates that universities should offer proactive interventions to improve the mental health of university students. The Charter suggests that targeted interventions are required for specific student groups, such as international students, as they may have particular needs, or may not access supportive services, such as counselling [26]. Consideration must also be given to the impact of wider academic determinants on students’ mental health and well-being, such as the availability of practical and logistical supports. However, there are currently no evidence-based guidelines on how to best support international student well-being, particularly in the context of the COVID-19 pandemic.

The aim of this study was to explore international students’ perceptions and experiences of the COVID-19 pandemic, social restrictions, and periods of self-isolation while registered for study at a UK university, and explore any potential impacts on their well-being and support needs. The findings of the study will inform policymakers and university authorities on how to best support international students during a pandemic.

## 2. Methods

### 2.1. Study Design, Participants, and Sampling

A qualitative study with focus groups and interviews was conducted with international students and employees at a university in central England. Participants included international students who were registered for study at the participating institution during the COVID-19 pandemic, and university employees (referred to herein as “staff”) who were in contact with international students during this time. Staff members who had a student-facing role, with particular emphasis on supporting international students, were invited to take part to gain a holistic view of the challenges faced by students during the pandemic. Focus groups with international student participants were conducted to facilitate discussions of differing experiences and perspectives [27]. A total of eight online focus groups were conducted, comprising 29 international students, with group sizes varying between three to five students. Seventeen one-to-one interviews were held with university staff members. We aligned our recruitment strategies with prior research which indicates that small focus groups are preferred by students to individual interviews or larger groups [28], provide in-depth insights, and foster a deeper understanding of participants’ experiences [29]. Individual semi-structured interviews were conducted with staff both for practical reasons of data collection during term-time, and to encourage open conversation around issues which may have been sensitive to discuss in a group setting [29].

A sampling guide was followed to maximise coherence, transparency, impact, and trustworthiness [30]. This included defining the inclusion/exclusion criteria (defining the sample universe); deciding on sample size; selecting sampling strategy and sample sourcing. Inclusion criteria were students studying abroad, who were registered for study at the participating organisation during the recruitment period. Home students were excluded from the study. We aimed to recruit a culturally heterogeneous sample. Purposive sampling was used to provide views from international students, using mixed-gender minimum quota sampling to recruit students with different nationalities, living circumstances (on/off campus), and experiences of COVID-19 (i.e., those who had recently tested positive for COVID-19, or never tested positive), and specifically targeted students who had experienced at least one period of self-isolation. Total sample size was informed by practical considerations (e.g., a pre-determined recruitment window of four weeks), since this was a rapid study intended to inform higher education policy and practice during a pandemic. The sampling criteria were based on published evidence. Students of different nationalities may vary in their experiences, needs [17,18], and access to services [26]. Students living on-campus may have easier access to services and lower stress levels than those living off-campus [31]. Finally, experiences of COVID-19 (i.e., having tested positive or not, having self-isolated or not) appear to have differential effects on psychological well-being [28].

Students required to self-isolate were: i) those that came to study in the UK from abroad and left the UK to go back to their home country, ii) those who tested positive for COVID-19 (with or without symptoms), iii) those who lived with someone who had COVID-19 symptoms or had tested positive. University staff members were purposively sampled for individual interviews, to represent the diversity of occupational roles and natures of student contact. Specifically, staff were invited to participate if they held a role providing direct support for international students, provided accommodation support, mental health support or other student-facing services, or worked in another role with a responsibility for pastoral care (i.e., academic tutor or supervisor). The study design aligned with the consolidated criteria for reporting qualitative studies (COREQ-32) guidelines [32] (Appendix A). The research protocol was approved by the University of Nottingham Faculty of Medicine and Health Sciences Research Ethics Committee (Ref: FMHS 96-0920).

### 2.2. Procedure

Data collection was carried out between January and February 2021, when stay-at-home restrictions and social distancing rules were in place in the UK, and students were arriving back to the university after a winter break. At this time, international students who had been away from the UK during the break were required to quarantine for 14 days upon arrival in the country. In this study, there was a mixture of international students who had recently travelled and experienced self-isolation, and those who had stayed in either the UK or in their home country. Recruitment occurred through two channels, firstly a subset of international students who had taken part in an established cohort study, involving a total pool of 897 students [33], and secondly, international students who had taken part in the university’s COVID-19 Asymptomatic Testing Service and had consented to be contacted for future research, involving a total pool of 133 students. Students meeting the sampling criteria were identified by the cohort leads and service administrators, and were subsequently contacted via email by a study researcher.

Student focus group participants were given a £20 voucher to thank them for their time. Staff members were invited by emails to participate in online semi-structured interviews. Students and staff were given access to study information, and those who were interested provided their consent online through the Jisc online survey platform. For practical reasons, recruitment continued for four weeks, in line with project timescales.

The research team conducted all student focus groups and staff interviews online using video-conference facilities (Microsoft Teams). Topic guides (Appendix A) were developed for both the focus groups and interviews by two psychologists (HB and HK), and were reviewed by five international students who were not part of the study team, to check understandability to those for whom English is not the first language. Key topics included in this guide were: exploring students’ experiences of self-isolation during COVID-19; the challenges that they faced; fears and concerns; and identifying facilitators, support, and coping mechanisms. Items about views towards vaccination were included, which will be explored in more depth elsewhere. The interviews took place at a time convenient to the participants. All interviews were conducted in the English language, and all participants were proficient in the English language as a criterion for study at the institution. Interviewees were encouraged to speak up if they did not understand the line of questioning; similarly, focus group moderators used prompts, and repeated or adapted questions if participants’ answers indicated they had not understood the essence of the question. Three researchers moderated the focus groups and conducted the semi-structured interviews and analysed the data. All interviews were recorded and transcribed verbatim. The focus groups with international students were led by two researchers who had personal experience of studying abroad (at undergraduate and postgraduate level, in the UK). One of these researchers was Turkish, and the other was Jordanian. Focus groups were co-moderated, and staff interviews were conducted by an English researcher, who had experience as both a student (postgraduate) and staff member (researcher/project worker) at the participating organisation.

Data were analysed thematically following Braun and Clarke’s six stages [34]. Three researchers read the transcripts multiple times to become immersed in the data until the data content was eventually known. Initially, focus groups and one-to-one interviews were reviewed separately. The initial coding for the staff transcripts was led by two researchers, and the students’ transcripts by a single researcher. Then, roles were exchanged where each researcher independently examined the coding for the students’ and staff transcripts to ensure reliability. This allowed the identification of commonalities and differences within the data sets. The potential themes were then generated by combining both data sets. This triangulation of the data provided a stronger understanding of self-isolation experiences, as the staff interviews enhanced the interpretation of the focus group data.

## 3. Results

### 3.1. Participant Characteristics

Eight focus groups, comprising between three to five students per group, were conducted with 29 international students in total (i.e., 16 were recruited from the university’s COVID-19 Asymptomatic Testing Service, and 13 from those who had taken part in the cohort study [33]). Seventeen university staff members were interviewed. The length of focus groups ranged from 58 to 90 min (mean: 68.8 min), whereas staff interviews were 29 to 69 min (mean: 44.5 min). The mean age of the students was 22.8 years (range between 19–32 years). Characteristics of the student and staff participants are displayed in Table 1 and Table 2, respectively.

### 3.2. Key Themes

Three key themes (Figure 1) were identified from the data analysis, with six subthemes: (1) practical, academic, and psychological challenges faced during self-isolation and the COVID-19 pandemic; (2) coping strategies to self-isolation and life during the pandemic; and 3) views on further support needed for international students. Themes and subthemes are presented below. A description of each theme and subtheme, including representative quotes, is presented in Appendix A.

#### 3.2.1. Theme 1: Practical, Academic, and Psychological Challenges Faced during Self-Isolation and the COVID-19 Pandemic

##### Sub-Theme 1: Logistics of Self-Isolation

In this study, there was a mixture of international students who were new to the UK, and those who had resided in the UK before. For those who were new, or returning to the UK, some described the challenges and fears associated with moving to a new country during the pandemic. Students travelling to the UK were required to self-isolate for two weeks, which led to difficulties setting up bank accounts, obtaining university cards, getting food, and other administrative tasks related to moving countries. The logistics of self-isolation posed unique challenges for many international students, particularly for those living alone or without existing social connections established in the community. Students spoke of difficulties getting essential supplies required for self-isolation, which led to a few students breaking self-isolation rules to meet their basic needs, highlighting the perceived lack of support for international students in self-isolation. For one student, the lack of practical support during self-isolation resulted in a negative impact on their physical health, as they could not obtain required medication.


*“I tested positive-I myself can’t go and I don’t have anyone that can pick it up for me and they didn’t find a way to send it to my house, so for those two weeks I could not—I didn’t have my medication.”*
(Focus group 8, student 5)


*“I think the food—we had a few students who were really worried about food, because they were all self-isolating.”*
(Staff member 11)

##### Sub-Theme 2: Transition to Online Learning

The transition to online learning following the pandemic was a cause for concern among international students. Students spoke of challenges related to adapting to using unfamiliar online platforms, with some reporting low knowledge of using such online platforms, and poor internet connectivity. Experiencing self-isolation left some students feeling unproductive, which, in turn, caused stress for some, as they felt behind in their academic studies. For those who had travelled to their home countries, connecting to live online lectures was a challenge due to differing time zones. A cause for concern for many was the uncertainties relating to the impact of the pandemic on their assignments for taught modules, or on field work, including laboratory sessions or placements. Some students reported frustration in the lack of clear communication from the university on COVID-19-related changes, stating it was a cause of stress and anxiety. For some, this stress and anxiety was further exacerbated by being signposted to mental health services when they perceived the problems to be related to other issues, such as a lack of clarity regarding academic processes and procedures, or a lack of clear communication from the university.


*“There’s just like no plans in place and no communication from the school. At the time we weren’t sure whether I should go back to Malaysia or not just in case we would still have to go back to the UK for the June exams and everything. There was just no clear communication and that was really worrying.”*
(Focus group 3, student 3)


*“I think there’s no point in giving like a lot of mental health support if the main cause of our mental health worries are about missing placements or not being able to get the most out of the teaching.”*
(Focus group 3, student 2)

Though some students spoke of the practical challenges of the transition to online learning, others felt that online teaching was not as valuable as in-person teaching, as they perceived that some courses needed the “practical element” of learning, including healthcare, engineering, and architecture courses. Being taught solely online caused resentment for some individuals, related to the fact that international students were paying much higher fees than home students, and perceived lower value from online lessons or supervision.


*“I’m paying double of what local students are paying, and not having lectures. Then on top of that, now I have to do everything online, I feel like the fees could have been reduced to lighten the burden.”*
(Focus group 2, student 2)

This view was mirrored by university staff, who advocated that online-based learning was not as effective as face-to-face lectures, particularly for international students where English is their second language, and that students may not know how to ask questions to lecturers through online platforms, potentially impacting their learning.


*“With English as a second language, it’s challenging enough to be writing essays, let alone thinking what is it that I’m actually asking? You can’t just write it in email, ‘I don’t understand’. It’s much easier to have those, just to verbalise things like that… for international students, I’m noticing that it’s harder, it’s just everything is harder.”*
(Staff member 9)

Furthermore, staff members suggested that online university courses undertaken in international institutions can be viewed less favourably than face-to-face teaching, which may impact on future employability. This was viewed by staff members as a cause for stress among international students, as it was perceived that students feared their degree would not be valued in their home country if it was conducted fully online, and that this was causing anxieties related to their future education and career prospects.


*“Some of the students have talked to me about not wanting to study remotely because the perception overseas is that the degree is not worth the same. So, it’s not worth coming to university because they don’t receive an online degree to be of the same quality as face-to-face teaching. I just, I’m not sure whether tied into that there are cultural perceptions about reaching out, whether it should be, you know, whether it’s still perceived to be a weakness, whether there’s still some stigma about it.”*
(Staff member 14)

##### Sub-Theme 3: Social Isolation, Loneliness, and Impacted Well-Being

Most students believed that self-isolation was necessary to prevent the spread of the virus; however, and as agreed by university staff, around half experienced feelings of loneliness, anxiety, worry, sadness, and low mood when they learnt they had to self-isolate. During self-isolation, international students (as with all students) could not engage in activities that would normally help them to de-stress, such as exercise, and seeing friends and family; this significantly affected their well-being. In contrast, there were some international students who viewed self-isolation in a more positive light, as it allowed time to engage in new hobbies, talk to family and friends by phone or online, and fostered a sense of gratitude once self-isolation had ended.

Staff highlighted the importance of “student connectivity”, in that international students need to immerse themselves in a new environment to fit in, which was not possible during this time, and the lack of social connections may have worsened feelings of loneliness and social isolation that are often experienced by the international community in more normal times.


*“I think mental health was like my biggest challenge. It was very easy to just feel down and not wanting to do things, not feel motivated to either do work or just get out of bed.”*
(Focus group 8, student 5)


*“I believe for our international community to arrive here into the UK and to find other people to talk to, to go to those societies… is part of university life. And that has been almost impossible for that to happen. So, you add the connection problems to the isolation problems that begins to impact on individual’s health and mental health.”*
(Staff member 8)


*“I think students often struggle because if you come to a foreign country where you don’t know anyone, like you can’t meet anyone. You know, like we had events for [students] in September or beginning of October for the global bodies, and I didn’t expect that, but we had over 100 students, which was really good! But it was just like it was me and one hundred students on teams… like there was no like conversations that we could, or they could have.”*
(Staff member 1)

Many international students highlighted a perceived lack of communication from the university during self-isolation periods and through the pandemic more generally, which left them feeling socially isolated from the rest of the university community. Some students reported little or no contact from university administrators, personal tutors, or lecturers. A perceived lack of contact from the university, compounded by the general reductions in social connection, led to feelings of disconnect from the university.


*“I didn’t exactly feel welcomed back when I came back after that. Compared to the first three years when I was at university, I thought it was great. The experience was great, staff was great. Now it just felt like it was deserted.”*
(Focus group 4, student 3)

In marked contrast, other international students spoke of how university staff reached out to them during periods of self-isolation, and through the pandemic more generally, which helped them to feel supported during this difficult time. Some students found that the level of pastoral care provided by university staff helped them to manage increased workloads arising from the transition to online working, and to manage elevated stress levels experienced during the COVID-19 pandemic.


*“My course has a course rep or ‘representative’, she would constantly send us survey forms on how we are doing mentally and how we are feeling with the changes. Like online schools, is there any improvements needed and how we feel, whether we feel stressed…. Applying for jobs is very intense and very stressful, so keeping up how we feel and letting us give feedback and suggestions anonymously really helped us.”*
(Focus group 4, student 1)

#### 3.2.2. Theme 2: Coping Strategies during Self-Isolation and Life in the Pandemic

Although periods of self-isolation posed some challenges, most international students used social and physical activities as coping strategies for self-isolating and living through the pandemic generally. Activities such as watching movies, chatting with family members and friends, reading, doing home exercises, and learning new hobbies were found to be helpful during the self-isolation period. These sorts of distractions were used as positive coping strategies for dealing with self-isolation and living under COVID-19 restrictions, and they made the isolation period feel shorter and were viewed to have a positive impact on mental well-being.


*“I think distraction is one of the most important things, distractions such as just being excited about the things that you can do at home like cooking so many foods and baking and then watching movies or like having the time to study, to focus, things like that. So, distraction from the pressure of just being alone, those balance out and helps mood in the process.”*
(Focus group 1, student 3)


*“So for instance, there are some online yoga courses that we are supporting some other courses like they directly contact through the computers just online courses. It’s kind of like social courses, they can join, and they can keep their life a little bit more active than just lying on the bed and then feeling they are sick and they’re dying. And I think that could be the only things that we can support with these students.”*
(Staff member 2)

Some staff highlighted that some students had not engaged in extra-curricular activities, and that a lack of distractions could be a source for stress and anxiety. Staff viewed international students to be highly driven to succeed in their studies, and witnessed students becoming more work-orientated during the pandemic, which served to intensify existing issues of stress and anxiety related to their academic studies.


*“I mean even pre-pandemic, the students that I was meeting were quite often having issues with anxiety. Stress about work specifically, like academic prowess and how good they were doing, and it was very much work focused. Now I think that’s been intensified, definitely because they have met so many less distractions.”*
(Staff member 10)

In the absence of physical events, online activities were viewed positively by most of the students as a chance to learn something new, engage in fitness classes, or meet new people. In contrast, staff highlighted that meeting new people and establishing relationships can be daunting for international students, and with the current COVID-19 restrictions, meeting new people may be extremely challenging and impede the international student experience. In normal conditions, staff spoke of student communities that form within specific international student groups, as people tend to bond with others from similar cultures and places; they observed that this community bonding was reduced or lacking during the pandemic. However, both staff and students viewed online activities as a potential avenue to help facilitate networking among students, in the absence of face-to-face activities, and many staff and students advocated for online activities to be better utilised and advertised.


*“Online social events could help students, they would not feel alone at this time, it would be very helpful. I know Students’ Union event team has been organising some things, but it hasn’t been well communicated and I don’t think that many people are aware of them, so yeah, I think more some sort of community and event would be very helpful.”*
(Focus group 7, Student 5)

#### 3.2.3. Theme 3: Support Needs from Institutions

##### Sub-Theme 1: Individual “Check-Ins” with Students

International students valued university staff reaching out to them, as it gave them a sense of being remembered and cared for, during periods when they felt particularly lonely. They specifically mentioned that individualised attention and personalised support was quite helpful during the self-isolation period. Staff views aligned with those of the students, and they reported that regular contact with students was beneficial to students’ well-being, to reassure students that staff were present and available to provide support if needed.


*“It really helps to have your personal tutor check in on you more frequently during this period, so my personal tutor he got in touch with me …., asked me if I needed any support. He asked me when I would be self-isolating …as well to ask me how I was doing. So, I think just having that sort of individualised attention quite helped because I knew that I had someone to turn to if I needed any support.”*
(Focus group 2, student 3)


*“I think it’s just checking in with them would be just quite helpful, just to say you know how you are doing, how you getting on, that frequent outreach or checking in with them would be quite useful to them because I think it just makes them think that OK, I know people are there, you know, they care about me.”*
(Staff member 3)

Nevertheless, students’ experiences appeared to have been highly variable, and some expressed disappointment and voiced dissatisfaction at the lack of support they received, particularly during self-isolation.


*“I think they could check on people, like, more individually too, because there was like feeling really left aside, yeah, like the feedback and checking more individually, I think that would be amazing.”*
(Focus group 7, student 1)

##### Sub-Theme 2: Improved Academic Support and Communication

Overall, international students called for improved communication and practical support from the university during the pandemic. The transition to online learning resulted in feelings of anxiety and stress for all, and students wanted to feel reassured that their education and future careers would not be impacted by the pandemic. They called for “safety net” procedures, which would protect their current grades and give “peace of mind” regarding the integrity of their degree.


*“I think for me it’s more-it’s just like more peace of mind about my education… there still wasn’t anything to support us, like, education, academically sorry, in this pandemic. So that’s one thing that would have eased me a lot, yeah.”*
(Focus group 7, student 2)

Timely information and decisions on how courses were being affected due to COVID-19 were seen as crucial by students in supporting their mental well-being during the pandemic, as this was a significant cause of anxiety, especially in times when some international students wanted to return home, but were unsure if they were able to during local and/or national restrictions. The lack of communication around COVID-19 impacts to courses led to confusion, and prevented many international students from returning home, which was viewed as crucial for well-being during the pandemic, since family and friends were key sources of support.


*“Support that I think would be helpful, just communication from [my] school, just knowing how going into self-isolation would impact my course, like how I’m going to progress or things like that, how missing out in classes or maybe placements would impact how I graduate and things like that. I think that would have been really helpful.”*
(Focus group 3, student 4)


*“Because for me, if they’re going to move everything online there’s not much of a need for me to stay here and especially with the current situation, there might not even be a graduation ceremony which I would stay for. It’s putting me in a very awkward place where do I go back, do I not?”*
(Focus Group 6, Student 1)

Staff recognised the importance of regular contact for international students, not only for them to feel connected to the university, but also for their English language to develop through daily communications and exposure to the language and culture.


*“Maybe some online activities that specifically focus on English practice, like language practice. I’ve had a few students who would like that because if they are self-isolating and stuff as well, they’re not having the opportunity to even go to the shop, and yeah, they can’t go up to someone on the street and practice asking for directions and things like that. So they are missing out on the everyday small interactions that they would normally be having.”*
(Staff member 10)


*“… I think, especially with PhD students or PGR students, one of the advice we give to our supervisors is do make it a weekly event that you’re meeting them online, so that they can, and then if necessary do the smaller one to ones or individual analysis and things like that and do not forget them in all the other things you are doing.”*
(Staff member 5)

##### Sub-Theme 3: Inclusive Mental Health Support

Despite many international students experiencing negative emotions at some point during self-isolation or the pandemic more broadly, most students stated that they did not access mental health support services. Some of the students appeared to be aware of the university’s services available to them; however, these international students described that, in situations where they were struggling with their mental health, they would talk to their friends about it, rather than seek formal support.


*“I didn’t really seek any support. Yeah, I did not feel like I needed the counselling services that the uni offers because I didn’t feel like it was a specific problem, I just didn’t feel motivated. I did try meditation, which is something that many people have suggested, and I think a support that really helps is just friendship, being able to talk to friends, facetime, or just chat.”*
(Focus group 7, student 5)

Although these student participants had not sought out mental health services, university staff stressed the importance of international students being aware of mental health services and using them if needed. Staff reported a perception that international students tended to load themselves with pressure to do well at university, often resulting from high family expectations. Staff highlighted that this self-imposed pressure may have been exacerbated during the pandemic, with the uncertainty around assignments causing more anxiety.


*“Kind of with this whole weight of, I can’t let my parents down, they’re paying loads of money, they’ve saved up all their lives. Maybe grandad and uncle and other people are contributing towards it, and so they’re kind of background of anxiety, like their resting level of anxiety is higher anyway, yeah, and then they’ve got to isolate and they’re in a foreign country.”*
(Staff member 9)

Staff believed that international students were less likely than domestic students to seek mental health services because of the stigma associated with it. University staff described mental health as a “real taboo” among international students, and had noticed that students were frequently hesitant to seek mental health assistance when it was needed.


*“A lot of international students can have mental health is still kind of big taboo subjects. They don’t necessarily, they’re not used to maybe talking about it openly, which is one of the reasons they might not reach out to someone.”*
(Staff member 15)

Interestingly, staff members spoke about the impact of the COVID-19 pandemic on their own workloads, which had challenged them emotionally, affected their own well-being, and, in turn, limited their ability to provide support to students.


*“Staff members had a really tough time in this pandemic. They did everything online…don’t know what’s coming next. And the workload essentially quadrupled. So, the problem is when, and they’re also human. We also go through all the emotional things students are going through. So, when you are not able to look after yourself, it becomes even harder to provide support for that many other students.”*
(Staff member 6)

## 4. Discussion

This study provides a qualitative exploration of the impacts of COVID-19 and social restrictions on international students. Research has largely focused on domestic students, with little attention given to the well-being issues of students from abroad who are registered to study in the UK. We present the psychological, social, and academic challenges faced by international students, potential individual and institutional implications, and offer recommendations for enhanced student support.

Our findings align with previous research that explores the impact of COVID-19 social restrictions on university students in a UK higher education setting [10,11]. It has previously been suggested that the practical, social, and emotional support needs of self-isolating students should be identified and should consider the needs of marginalised groups [11]; although, to date, little is known about the experiences of international students as a unique population.

The rapid transition to online learning during the pandemic presented unique challenges for international students. Unfamiliar online learning environments led to difficulties articulating educational issues and queries in a second language, which may have limited access to previously received support. For those studying in their home countries, additional challenges included poor internet connections and studying in different time-zones. Available literature shows that the speed of adaptation to online leaning is highly dependent on the students’ psychological and technological ability [36], and that difficulties adapting to online learning may trigger high academic distress, increased stress, anxiety, and depression [24,37]. Some students noted that these structural difficulties impacted their mental health to a greater extent than other pandemic factors.

Research suggests international students experience high degrees of perceived isolation, both academically and socially; however, students studying exclusively online may experience greater isolation than those completing traditional face-to-face programmes [37]. Students struggled to adapt to online learning due to language and technology barriers, and needed greater support from their institution in transiting to this learning platform, which may have heightened their stress and anxiety during this time. Universities need to ensure that international students have ongoing access to technical and logistical support for navigating remote learning environments, including (and in particular) those who continue their education remotely, accessing online learning platforms from their home country. As some UK institutions may choose to retain elements of online teaching and learning within their traditional teaching curricula, consideration should be given to supporting incoming international students as they transition to using online platforms, for example, through introductory courses prior to, or immediately when starting, their course of study.

A common thread that emerged throughout the discussions was the need for greater communication from the university during the pandemic, including the provision of information about course alterations and regular check-ins by university staff. A perceived lack of communication resulted in increased anxiety amongst some of the international students in our sample, whereas increased pastoral communication improved connectedness. During the COVID-19 pandemic, communication was paramount to the uptake of COVID-19 testing and protective behaviours in students [38]. Delays in communication and a lack of messaging clarity caused confusion and impeded behavioural adherence [38]. For international students specifically, clarity and accessibility in both the frequency of and language used in university communications may reduce feelings of anxiety, particularly in individuals for whom English is not first language. Students within our sample described concerns that their overall performance, future career prospects, and quality of their degree would be impacted by the shift to online learning—many were concerned that this topic was not sufficiently covered within university communications. To attract and retain new students, information regarding the impact of online learning on course outcomes should be made readily available. The standards of online learning should be communicated clearly to international students (and their sponsors) to alleviate their concerns related to future career prospects of online study versus face-to-face modalities. Universities should provide timely communication to international students (and indeed, all students) about changes to courses, procedures, and service availability. This ideally would be available in alternative formats (e.g., online discussions) to avoid full reliance on lengthy, written, email communications, which can be challenging for students who are not native English speakers. Furthermore, to provide effective and focused educational support, universities must consider a holistic approach, through the assessment of international students’ specific concerns and needs, including cultural, linguistic, and social factors.

Emerging evidence suggests that international student mobility has drastically decreased during the pandemic, with many international students advised not to travel abroad for study [38]. Beyond student enrolment, the impacts of the pandemic on student transition and integration may also have implications for academic attainment and cultural immersion. International student integration into a new culture significantly predicts second-language proficiency, in turn, predicting successful academic adaptation [39]. For international students, moving abroad and adjusting to a different country’s culture and norms can pose significant challenges that affect mental health [40]. Additional evidence suggests that international students who returned to their home countries during the pandemic have been deprived of access to the cultural knowledge of the destination country, as well as of the insights typically arising from face-to-face teaching and social interactions [40]. Our findings suggest that simple mechanisms, such as pastoral communication, helped to improve student connectedness with their university courses, and may, in turn, foster cultural adaptation. Furthermore, the role of online social connections was also viewed as important for student well-being by both staff and students, including online exercise classes, conversations with family and friends, as well as events organised at a university level. The literature suggests that social connections are particularly important psychosocial resources for international students adapting to life in a new country [41]. International students share similar characteristics, regardless of their cultural and ethnic background, and are, therefore, more likely to form social connections with other international students [42]. Forming contacts with students from similar national groups allows for the exchange of experiences, and establishes a supportive network that improves well-being [43]. However, opportunities to make connections with home and international students have been limited due to COVID-19 restrictions. Despite the limited opportunities to establish supportive networks face-to-face, both international students and staff recognised the utility of online platforms to facilitate social activities during periods of self-isolation and COVID-19 restrictions, which was reported to be beneficial to students’ psychological health and well-being. International students were still motivated to attempt to build social networks through online platforms, suggesting that online social events may allow increased opportunities to form social connections.

Difficulties with adaptation were likely compounded by periods of self-isolation during the pandemic, with some students in our sample coping well, whereas others experienced loneliness and feelings of detachment from university life. In line with these findings, recent studies reported deteriorations in general student mental health, including feelings of isolation, loneliness, concerns about catching the virus, stress, anxiety, and depressive symptoms [44].

Addressing mental health concerns in university students is imperative, with potential impacts on physical health and academic progression. For example, mental health is associated with increased susceptibility to disease—there is emerging evidence to suggest that psychological factors may influence susceptibility to SARS-CoV-2 infection [45]. Regarding academic outcomes, students with mental health concerns are twice as likely to leave an institution without graduating; they have lower Grade Point Averages (GPAs), enrol and drop out in intervals, take longer to earn their credentials, or drop out completely [46].

International students already face stressors related to studying abroad, including educational difficulties, loneliness, and practical problems associated with changing environments [41]. Although international students are more likely to experience issues with mental health, they are less likely than home students to seek help from the university [47]. Within some cultures, counselling or other forms of formal support for mental health difficulties remain stigmatized [43]. Therefore, current services offered by British universities, which rely on individualistic Western models in which students proactively seek support, may not be fit for purpose for certain student groups. Our findings highlight that international students tend to seek informal support from friends and family, rather than formal university support systems. Psychological support received through social networks, such as family and friends, has been linked to reduced feelings of depression and lower levels of loneliness [48,49,50]. Further, peer support may also moderate the relationship between acculturative stress and anxiety symptoms [51], in addition to improving academic achievement [52] and self-efficacy [53]. The role of peers in mental health support may be multifaceted, ranging from signposting of services to the provision of support services themselves. Early data from an online peer support initiative demonstrated that 71% of students engaging with this platform experienced some improvement in pandemic-related stress [54]. Given that international students demonstrate higher rates of mental health conditions than those of their home student counterparts, it is imperative that further research focuses on delivery modalities that align with the needs of international students, for example, by incorporating peer support or peer advocacy for service signposting. Students in this study saw online social events as a route to engaging with the university community, which was crucial for positive well-being; therefore, universities should promote online events aimed at international students to protect well-being. Proactive approaches from university staff may provide a starting point to encourage international students to discuss mental health and well-being more openly; however, further initiatives should be focused on breaking the taboo on mental health in international students, to ensure access to services, particularly during the transition into post-pandemic academic life. As higher education institutions become increasingly culturally diverse, the need for inclusive mental health support is paramount—this means ensuring that mental health services are meaningful and accessible to all students with varying values, religions, beliefs, and cultural backgrounds. Offering various mechanisms for psychological intervention, such as proactive approaches (“reaching out” to student communities) and online peer support schemes, may benefit all students. Therefore, offering inclusive mental health services will likely improve access to psychological support across the university, and, in turn, serve a wider population of students [55].

University staff well-being was found to be impacted in this study. Student-facing staff had experienced significant pressures during the pandemic due to increased workloads because of the transition to online working, leaving some staff experiencing work-related stress. Previous studies have found that university staff experienced stress during this time, as they adapted to home working and dealt with an increase in workload; re-gaining a sense of control seemed to be essential to reducing stress over time [56]. In this study, some staff highlighted feelings of guilt, as they believed their increased workloads prevented them from supporting students (i.e., being immediately available to them) to the best of their ability. Many university staff struggled with the increased and unique support needs of international students during the pandemic, alongside their usual workloads. Educational institutions should increase the availability and accessibility of centralised support for international students to improve the parity of support provided and reduce the burden on individual staff. Further mental health support for staff may also be warranted. A survey of 55 university staff found that 22–24% of participants reported clinical-level anxiety and depression scores, and 66.2% experienced high stress levels due to COVID-19 [45]. In this study, specific staff groups that required further support were highlighted, including staff members with care duties, and staff from minority ethnic backgrounds, who may be at increased risk with regards to the virus. University staff should have access to well-being and mental health support to avoid the risk of burnout during unprecedented circumstances such as a pandemic. If additional support is to be offered to students, this needs to be coupled with increased staff support and recruitment, to ensure staff workloads do not become excessive.

Implications for higher education policy and student support services are shown in Box 1.

Box 1Recommendations and strategies to minimise psychological impacts of a pandemic for international students.Recognise the significant practical and emotional impacts of a pandemic on international students, and account for this when assessing student engagement in studies and academic progress.Ensure equitable mobilisation of basic supplies for students living on and off campus, in the face of another pandemic.Acknowledge the additional challenges faced by students who may work remotely during a pandemic—such as poor internet connections and studying in different time-zones.Provide ongoing access to technical and logistical support for navigating remote learning environments, including (and in particular) those who continue their education remotely, accessing online learning platforms from their home country.Provide additional services to support incoming international students as they transition to using online platforms, for example, through introductory courses prior to, or immediately when starting, their course of study.Ensure timely and regular communications relating to the provision of information about course alterations. Use alternative formats for the dissemination of key information.Involve international students in reviewing and developing the student-facing materials and messaging distributed during a pandemic.Provide an increased level of pastoral support for students, including regular check-ins by university staff. The impact on staff workloads and well-being should be considered, and there should be increased investment in centralised support services where appropriate.Seek to enhance social connectedness, inclusion, and positive mental well-being through supportive services delivered through various mechanisms (face-to-face, online, through local communities).Increase opportunities for international students to connect with others from their cultural and community groups.Endeavour to reduce perceived stigma among international students, related to accessing counselling services, or other forms of formal support for mental health difficulties.Encourage access to informal support (friends, family, community groups).Initiate service-led drop-in sessions for student queries and signposting.Provide peer mentoring or “buddy” programmes for international students—train and support peer mentors/buddies in signposting to supportive services.Invest in support and welfare services to prepare for the longer-term impact of pandemic-related mental ill-health on international (and all) students.

### Study Strengths and Limitations

To date, there has been limited qualitative exploration into the impact of the COVID-19 pandemic on international university students. This study has demonstrated that international students’ well-being has been impacted by the pandemic, and that individuals may face different challenges to domestic students. One of the key strengths of this study is looking at the perspectives of students from a range of ethnic and cultural groups, in conjunction with views from staff members, which broadened our understanding of international students’ experiences with self-isolation, and highlighted the “hidden” challenges affecting their well-being. Our student sample was heterogeneous, which is of value in cross-cultural qualitative research, since any commonality found across a diverse group of cases is more likely to be a widely generalisable phenomenon than a commonality found in a homogenous group of cases (i.e., our findings are not confined to a particular cultural group, or stage of study) [30]. Online focus groups and interviews allowed us to reach students and staff both on- and off-campus without encountering logistical challenges or costs. However, the face-to-face approach may be valuable in recruiting hard-to-reach or vulnerable populations, as it builds trust [30]. Our sample included more women than men, which aligns with long-standing evidence that women are more likely to self-disclose than men [57]. Our findings are limited to the views and experiences of international students studying in the UK. It is recognised that the students who took part in the focus groups were likely to be proactive and engaged, and may not represent the views of those who felt more disconnected from the university or were less comfortable talking in this setting. We did not collect data on the type of degree that the students were registered for, and there may be differences in students’ experiences during the pandemic. For example, prior research has suggested that students’ willingness to discuss mental well-being (or other personal challenges), and the level of pastoral support they receive, can vary across disciplines [13]. Further, there may be differences in experiences of students registered on taught programmes, compared with those on research degree programmes. We did not collect information on socioeconomic status, which is a known risk factor for mental health; therefore, further research may be conducted to fully understand the impacts of the pandemic on international students from lower income backgrounds [58].

## 5. Conclusions

This study enhances our understanding of the experiences and vulnerabilities of international students enrolled in a UK university during the COVID-19 pandemic. Our findings provide insights into how higher education institutions can best support international student groups during this pandemic. Primarily, this may be achieved through clear communication strategies with alternative formats for the dissemination of key information, regular contact and support, and the provision of proactive and tailored mental health services (such as peer support programmes and drop-in sessions) accessible to international students. Universities should actively encourage the formation of online social communities and events for international students to encourage immersion into university environments, and provide additional practical, academic, and emotional support to international students adapting to online learning environments. International students enhance the overall student experience and students’ personal development. The UK must respond to the impacts from the pandemic on international students, to retain and expand international student recruitment in the future.

## Figures and Tables

**Figure 1 ijerph-19-07631-f001:**
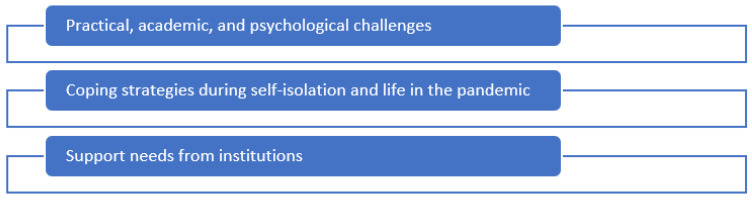
Key themes identified.

**Table 1 ijerph-19-07631-t001:** Characteristics of international students (*n* = 29).

		Participants *n* (%)
Gender	Male	8 (27.5)
	Female	21 (72.5)
Nationality *	Americas (Brazil, Chile, Mexico, other)	4 (13.8)
	Eastern Mediterranean (UEA)	1 (3.5)
	Europe (Belgium, Croatia, Turkey)	4 (13.8)
	South-East Asia (India, Indonesia, Thailand)	4 (13.8)
	Western Pacific (China, Malaysia, Singapore)	11 (37.9)
	Not specified	5 (17.2)
Degree level	Undergraduate	22 (75.8)
	Postgraduate	7 (24.2)
Type of accommodation	On-campus	7 (24.2)
	Off-campus	22 (75.8)
COVID-19 positive **	Yes	3 (10.3)
	No	26 (89.7)
Currently self-isolating	Yes	3 (10.3)
	No	26 (89.7)
Isolated more than once	Yes	12 (41.3)
	No	17 (58.7)

* Regions defined by World Health Organization definition of the world’s countries by continent [35]; ** Received a positive test result for Severe Acute Respiratory Syndrome Coronavirus-2 (SARS-CoV-2) RNA as per voluntary self-report.

**Table 2 ijerph-19-07631-t002:** Characteristics of university staff members (*n* = 17).

		Participants *n* (%)
Gender	Male	4 (23.8)
	Female	13 (76.2)
Role *	Health and well-being	5 (29.5)
	Accommodation support	1 (5.9)
	Teaching and academic support	7 (40.8)
	Student experience	4 (23.8)

* Health and well-being roles: mental health advisors and well-being managers; accommodation support roles: hall tutors; teaching and academic support role: academic staff and personal tutors; student experience support role: student services administrative staff members and careers staff.

## Data Availability

The data presented in this study are available on reasonable request from the corresponding author.

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
