# Peer review of "Exploring the Psychological Impacts of COVID-19 Social Restrictions on International University Students: A Qualitative Study"

_ijerph, 2022, doi:10.3390/ijerph19137631_

Round 1

Reviewer 1 Report

Review report

The aim of the paper is to offer an exploratory view on social life of international students, during the COVID pandemics. It main contribution is the highlight of specific themes/dimensions, identified based on qualitative methods (focus group and interview).

The cited references are mostly recent (2020, 2021, 2022) and relevant for the topic.

The qualitative methodology is appropriate for the topic.

For the procedure – it is not necessary to indicate which researcher did a specific part (for example lines 179, 183, 187)

Results

The presentation of the results could be more extensive – a proposal could be a differentiation between the results of the focus group and those from the interview – since there are two different participant groups.

In the discussion part, the results from the staff category are minimalistic presented, as well as the relation to the existing literature (qualitative and quantitative one).

An improvement of services for international students can be more explicit revealed, as well as the impact of the results of such a study on long term policies, services of the universities.

Reviewer 2 Report

Review for ijerph-1759465

Title: Exploring the psychological impacts of COVID-19 social restrictions on international university students: A qualitative study.

This interesting qualitative study explores the experiences of international university students during COVID-19 restrictions and their impact on different aspects of university life. Although the title of the paper suggests a focus on the psychological impacts of these restrictions, other themes, including practical and academic challenges, were identified and discussed. The paper is well written and gives a good overview of the challenges during the pandemic that are particularly relevant to international university students in the UK. I agree with the authors that this group of students has not been studied adequately and that results from this study can inform interventions and student support provided by higher education institutions. The introduction reads well, offers some very interesting figures, and sets the scene nicely for the aims of the study. The method section is clear and with good detail regarding design and procedure. Overall, the results are interesting, informative, and well presented, with relevant quotes from student and staff participants. The discussion summarises the key findings well, and makes good links with the existing literature and recommendations for good practice.

I have a few minor suggestions:

A bit more information on recruitment would be helpful. The authors report that recruitment occurred through two channels: a) a subgroup who had taken part in an established cohort study and b) another subgroup that had taken part in the university’s covid-19 asymptomatic testing service. I would welcome some more detail on how these two subgroups were identified and how many students were approached and agreed to take part in the study.

Characteristics of student participants: There is a small number of postgraduate students taking part in this study and I wondered whether these were all MSc/MA/Diploma or Research PhD students. The experiences of lock down can vary depending on what degree students were doing at the time of the research.

I would also like to know what courses the student participants (UG and PG) were doing. Different courses might have created different challenges for students (e.g., professional vs no professional degrees). I wonder if this is something worth mentioning in the paper as it can add further insight into these students’ experiences.

I found the discussion about the culture around seeking wellbeing and mental health support among international students interesting and it certainly reflects what the literature in this area is showing. Would it be possible to add a bit more information in the participants’ characteristics such as the ethnicity of student participants?

Finally, what do the authors mean by inclusive mental health support and why have they chosen this term? I would love to see some discussion, even if brief, about the importance of inclusive mental health support and why this is relevant to international university students.

Finally, it would be helpful to include some examples of questions used during the focus groups with students and interviews with staff for a clearer idea of the measures used in this study.

Reviewer 3 Report

The article is devoted to a topical issue - the mental health of young people, namely those who are particularly vulnerable in the Covid-19 pandemic. Various studies have shown that social isolation and restrictions in place to limit the spread of the virus negatively affect different populations, including young people. At the same time, international students have experienced particular difficulties related to educational issues, difficulties of adaptation, etc. On the one hand, students can be considered as a fairly homogeneous group - with high levels of physical health and psychological well-being. However, on the other hand, in research on the mental health of students during the pandemic, researchers do not focus on international students. They usually do not become a separate study group. The UK is known for its policy on students' mental health; universities have developed clear protocols to help with the declared signs of problems with mental health. At the same time, for the effective functioning of such protocols, it is important to understand the current state of affairs, the characteristics of the participants, and so on. In this context, in my opinion, the article is interesting and relevant.

The authors used high-quality research methods - this allowed them to obtain a large amount of data and identify and classify problems. Also, I can assume that quantitative research methods might be rather superficial in this case.

Among the weaknesses of this article is the insufficiently detailed description of the sample and the incomplete indication of limitations.

I think it is important in the methods to answer such questions - whether this sample can be considered as a representative, whether it is homogeneous in terms of culture and traditions (whether students come from similar/different cultures); in which language the interview was conducted, at what level the student speaks the language used during the interview; whether it was checked that the student understands the content and essence of the question.
